# Real-time tracking of metal nucleation via local perturbation of hydration layers

Robert L. Harniman[1], Daniela Plana [1,4], George H. Carter[1], Kieren A. Bradley[1,2,5], Mervyn J. Miles [3] & David J. Fermín [1,2]

The real-time visualization of stochastic nucleation events at electrode surfaces is one of the most complex challenges in electrochemical phase formation. The early stages of metal deposition on foreign substrates are characterized by a highly dynamic process in which nanoparticles nucleate and dissolve prior to reaching a critical size for deposition and growth. Here, high-speed non-contact lateral molecular force microscopy employing vertically oriented probes is utilized to explore the evolution of hydration layers at electrode surfaces with the unprecedented spatiotemporal resolution, and extremely low probe-surface inter-action forces required to avoid disruption or shielding the critical nucleus formation. To the best of our knowledge, stochastic nucleation events of nanoscale copper deposits are visualized in real time for the first time and a highly dynamic topographic environment prior to the formation of critical nuclei is unveiled, featuring formation/re-dissolution of nuclei, two-dimensional aggregation and nuclei growth.

---

[1] School of Chemistry, University of Bristol, Cantocks Close, Bristol BS8 1TS, UK. [2] Bristol Centre for Functional Nanomaterials, University of Bristol, Tyndall Avenue, Bristol BS8 1TL, UK. [3] School of Physics, H.H. Wills Physics Laboratory, University of Bristol, Tyndall Avenue, Bristol BS8 1TL, UK. [4]Present address: School of Chemical and Physical Sciences, Lennard-Jones Laboratories, University of Keele, Staffordshire ST55BG, UK. [5]Present address: Ilika Plc, Kenneth Dibben House, University of Southampton Science Park, Enterprise Road, Chilworth, Southampton SO16 7NS, UK. Robert L. Harniman and Daniela Plana contributed equally to this work. Correspondence and requests for materials should be addressed to D.J.F. (email: david.fermin@bristol.ac.uk)

Electrochemical deposition offers the most versatile and scalable route for material growth, spanning from atomic-scale control of sophisticated architectures to large-scale industrial processes (e.g. electrowinning)[1–7]. The mechanism of nucleation and growth of metallic phases has been mostly studied employing electrochemical transient measurements, for which various models have been developed linking the time-dependent current to microscopic parameters such as nucleation rate and nuclei density[8, 9]. However, discrete events during the early stages of electrochemical nucleation are notoriously difficult to probe in real-time, mainly due to their stochastic nature, fast mass transport phenomena to nanoscale sites and highly dynamic topographic changes. Indeed, the accuracy of classical electrochemical nucleation models based on current transient responses, particularly in the early stages of phase formation, has been called into question with the advent of powerful tools for nanoscale characterisation[10–17]. For instance, in-situ transmission electron microscopy revealed significant differences between the nucleation and growth of individual nuclei and that predicted by classic models; e.g. in early stages, nucleation rates are slower and nuclei densities are higher than expected[14, 18, 19]. Recently Mirkin and co-workers have been able to follow the nucleation of single particles at low overpotentials employing electrodes of 50 nm in diameter[20]. The stochastic nature of interfacial nucleation in such spatially confined regime manifests itself by phenomena such as induction times, i.e. period of time after the deposition potential is applied in which no stable nucleii are formed. Rationalising these phenomena from electrochemical data is highly complex due to the fact that no faradaic current is recorded prior to the formation of the critical nucleus Scanning probe techniques, including high spatially resolved spectroscopic techniques, have had a profound impact in understanding processes taking place at electrode surfaces[21]. Video-rate scanning tunneling microscopy (STM, ≤20 frames s$^{-1}$) under electrochemical environment has been used to investigate processes such as ion adsorption, metal dissolution and two dimensional growth with exquisite atomic resolution[22–30]. However, the kinetics of 3D phase formation can be substantially affected by so-called shielding effects linked to the substrate-tip potential difference[13, 31]. The close tip-substrate interaction in conventional scanning probe microscopy can severely disrupt the structure of the nascent metal deposits as demonstrated by the work by Penner and co-workers[32]. State-of-the-art AFM methods, as reviewed by Mandal and co-workers[33], have uncovered detailed images of particle growth in the sub-micron scale, while Unwin and co-workers have developed powerful new approaches to surface mapping, based on scanning electrochemical cell microscopy (SECCM), reaching up to 4 s per frame[34, 35]. This versatile technique enables the confinement of nucleation phenomena to regions as small as 500 nm, allowing the possibility of linking topographic features with preferential nucleation sites as well as particle nucleation and detachment[36]. However, visualizing individual stochastic nucleation events in real-time requires an entirely different approach with unprecedented spatio-temporal resolution.

Herein we describe an approach to in-situ real-time monitoring of stochastic nucleation events based on detecting local fluctuations of the hydration layer at an electrode surface employing high speed lateral molecular force microscopy (HS-LMFM), a non-contact technique, operating with an optical vertical feedback mechanism. Our approach features vertically-oriented probes (VOP) with spring constants as low as 30 fN nm$^{-1}$ [37], the position of which can be tracked, with sub-second temporal resolution[38], by scattering of the evanescent wave (SEW) of a laser undergoing total internal reflection at the back side of an optically transparent electrode. This configuration is capable of resolving, for example, individual stepping forces of

kinesin[39], the ultra-structure of self-assembling cages composed of peptide modules[40] and even the momentum promoted by polarized light perpendicular to the wavevector[41]. To the best of our knowledge, we monitor single copper nucleation events on In-doped SnO$_2$ electrodes (ITO) for the first time, opening a window into the complex stochastic processes characterizing the early stages of metal phase formation. While the Faradaic current exhibits small temporal changes, a rich variety of processes can be monitored, such as nuclei formation/dissolution, as well as two-dimensional nuclei aggregation and growth.

## Results

**High-speed lateral molecular force microscopy.** The distance between the VOP tip and the electrode surface is finely controlled as a result of the exponential decay of the evanescent field generated via total internal reflection from laser beam through a high, 1.49 NA, objective lens (Figs. 1a, b). When a VOP is tens of nanometres away from the substrate, the tip diffracts and scatters the non-propagating evanescent field, generating a propagating wave which is measured with a photo-detector[42]. It is therefore important for the substrate used to be transparent to the wavelength of the detection laser, in this case a thin coating of ITO on a glass coverslip was utilized. The hybrid nature of the microscope allows the force interactions between the VOP and hydration layers[43] to be collected simultaneously with, but independent of, the optical feedback signal[38]. The viscoelastic response of the water layers between tip and sample changes the resonant dynamics of the cantilever[43]. By oscillation of the VOP at or close to its resonant frequency, it is possible to measure changes in resonance dynamics via the scattered propagating light collected at the detector. The optical feedback mechanism has a lateral resolution comparable to the scattering area of the VOP and therefore detects only local changes in the evanescent field. An additional advantage of optical feedback is that the tip height is always maintained relative to the sample, meaning that any drift within the system, whether it be electrical drift in piezos or thermal drift in the mechanics, will not influence the experiment. Once immersed in the electrolyte, the VOP tip was positioned, with sub-nanometre vertical precision, above the ITO roughness level (~2.9 nm) at the separation where shear force interaction became negligible. As the metal deposits grow, their hydration layers (represented by concentric blue regions in Figs. 1a, c) move into the detectable range of the VOP and the increasing shear-force is detected in the form of a resonance frequency shift of the probe. As the probe is raster scanned at a constant height from the substrate, utilizing an X-Y stage with a resolution in each axis of 0.2 nm [P-734, Physik Instrumente, Germany], these frequency shifts are mapped on a pixel by pixel basis to generate a shear-force map Fig. 1c. Figure 1d is a representative force-retraction curve that exemplifies the change in probe oscillation with increased separation from the substrate, here in the case of a clean mica surface in ultra-pure water. The various steps, highlighted by dashed lines on the graph, correspond to the hydration layers schematically illustrated in Fig. 1a, typically between four and six layers. Sub-nanometre oscillation of the VOP tip ensures that forces generated parallel to the plane of the substrate are not larger than 20 pN, while the mapping of hydrodynamic interactions at a constant distance ensures negligible forces normal to the plane of the substrate.

**Copper nucleation kinetics.** Cu nucleation kinetics on ITO were investigated by chronoamperometry in order to establish the appropriate Cu$^{2+}$ concentration in solution and electrode potential for real-time visualization of nucleation events. The electrolytic bath consisted of an aqueous solution of CuSO$_4$ (pH 3)

with $Na_2SO_4$ as supporting electrolyte. Further experimental details can be found in the Methods section. A range of chronoamperometric steps recorded in the presence of $1.0 \times 10^{-3}$ mol $\times$ dm$^{-3}$ CuSO$_4$ at various overpotentials are shown in Supplementary Fig. 1. The chronoamperograms show the characteristic nucleation maximum which shifts towards shorter times as the overpotential is increased, followed by planar diffusion limiting current after 20 s. Dimensionless current transients analyzed

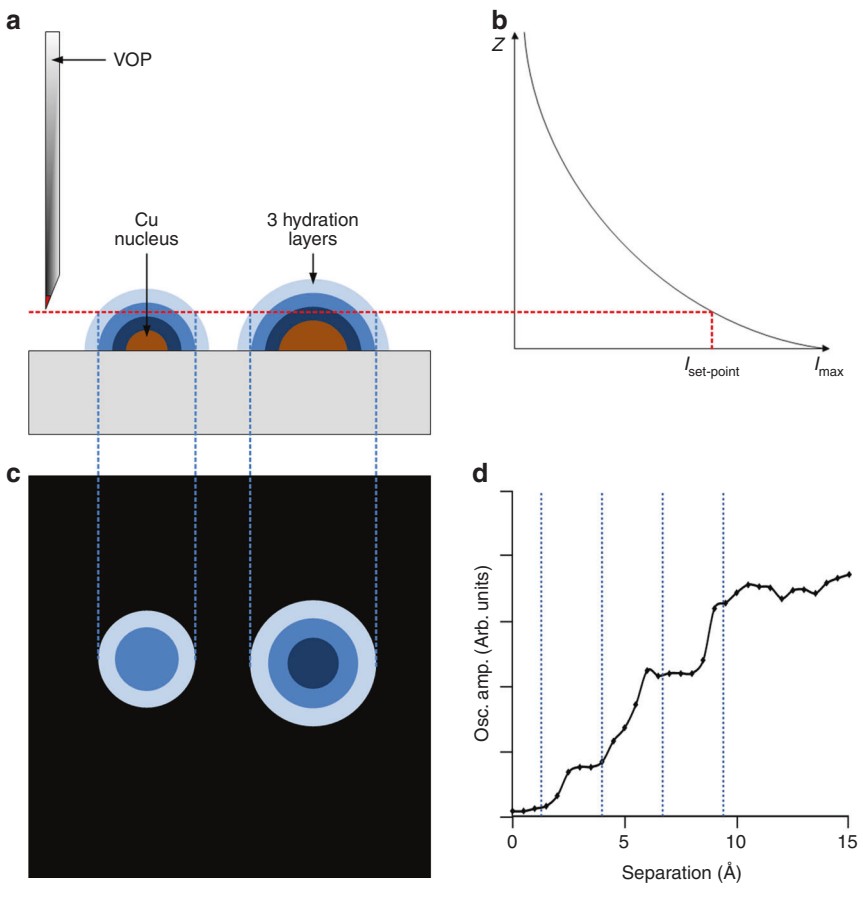

**Fig. 1** Schematic of the optical feedback detection of the hydrodynamic signatures of nucleation. **a** Vertically oriented probe (VOP) scanned above the nascent Cu nuclei on the ITO electrode (grey). The blue gradient represents the strength of shear-force interaction around the nuclei caused by hydration layers. The red dashed line shows the scatter intensity used as the set-point to keep the probe tip at a constant separation from the ITO slide surface. **b** The exponential decay of the evanescent field with the distance ($z$-axis) from the surface of the ITO and how this relates to the intensity set-point in panel **a**. **c** Schematic representation of the contrast (probe frequency shift) induced by the shear-force interaction experienced by the tip while scanned above evolving nuclei at a given set-point. **d** VOP oscillation amplitude as a function of distance from a clean mica surface in ultra-pure water. This curve illustrates the hydration layer interaction in HS-LMFM imaging

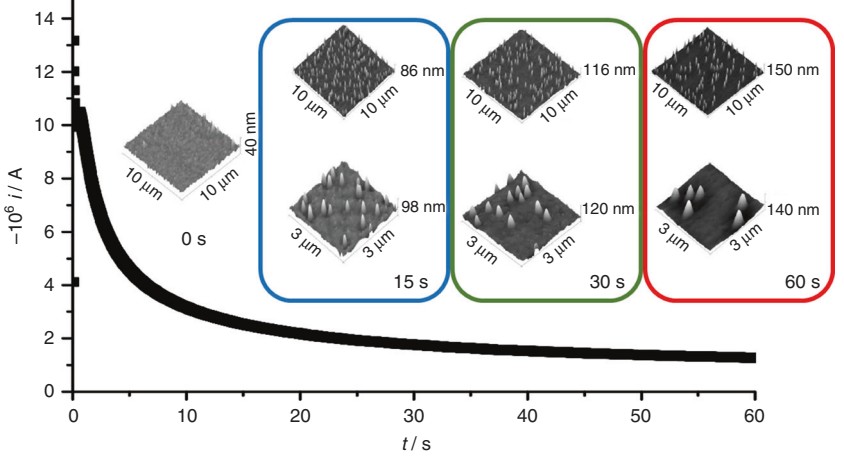

**Fig. 2** Current transient recorded at $-0.34$ V overpotential on an ITO electrode in $1.0 \times 10^{-3}$ mol dm$^{-3}$ CuSO$_{4(aq)}$. AFM images recorded ex-situ after 0 (ITO topography), 15, 30 and 60 s are superimposed in the figure. Statistical analysis of a number of electrodes resulted in a constant nuclei number density of $(1.5 \pm 0.8) \times 10^8$ cm$^{-2}$, which is consistent with an instantaneous nucleation mechanism

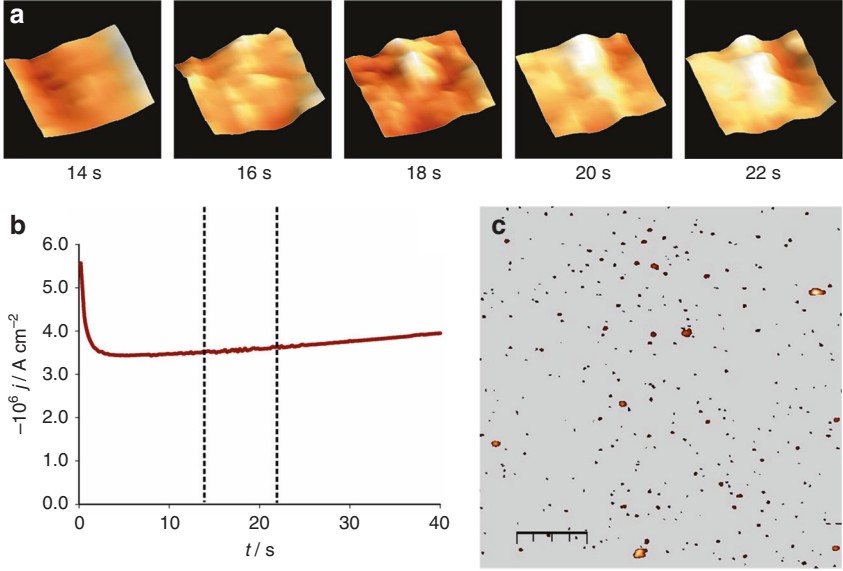

**Fig. 3** Birth and growth of a copper nucleus. **a** 46 × 46 nm² cropped regions from 85 × 85 nm² in-situ HS-LMFM scans, taken from 14 to 22 s after the potential step (*dashed lines* on **b**), highlighting the birth and growth of a copper nucleus. **b** The chronoamperometric transient recorded at −0.27 V in 1.0 × 10⁻⁴ mol × dm⁻³ CuSO₄ solution. **c** Ex-situ AFM image recorded after a 60 s potential step under the same conditions; the *scale bar* represents 550 nm

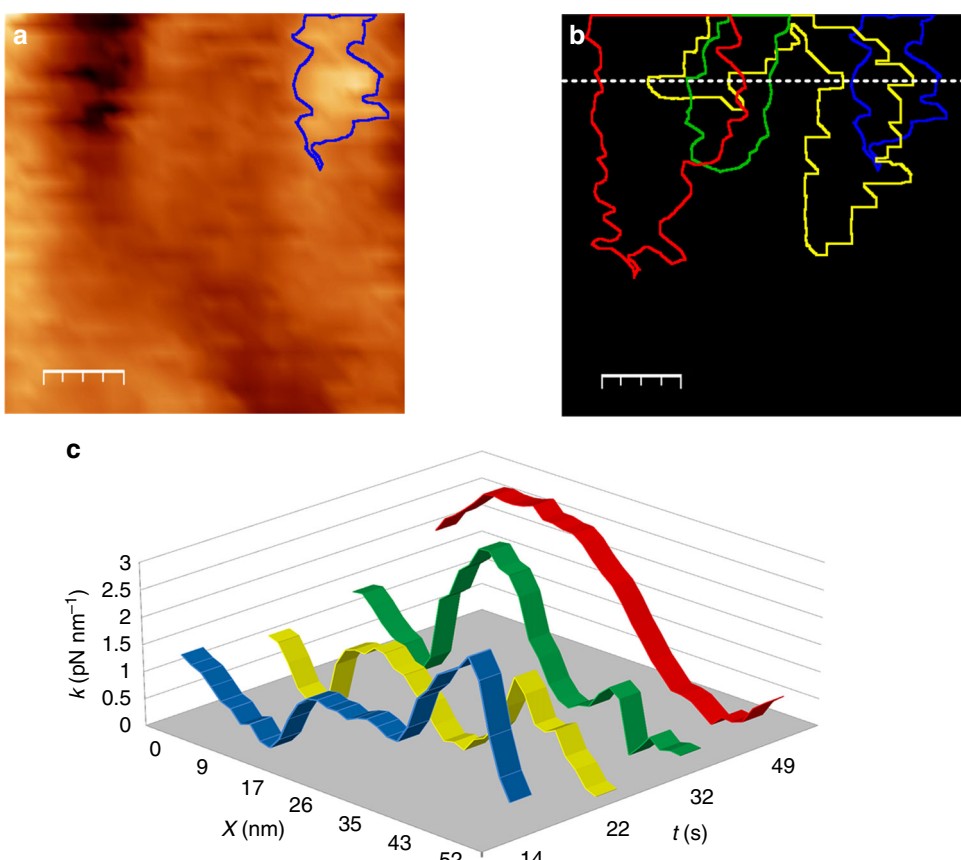

**Fig. 4** Dynamic growth of a Cu nucleus on the ITO surface at constant overpotential. The force micrograph at 14 s after the potential step is shown in **a**, where a copper nucleus is highlighted in *blue*. Contours of the copper features at 14, 22, 32 and 49 s are presented in **b**, while **c** shows the shear-force as a function of time and position (over the *dotted line* in **b**). The *scale bars* in **a** and **b** represent 12 nm

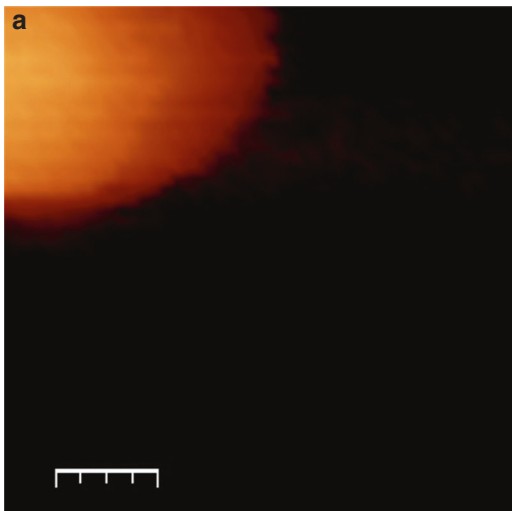

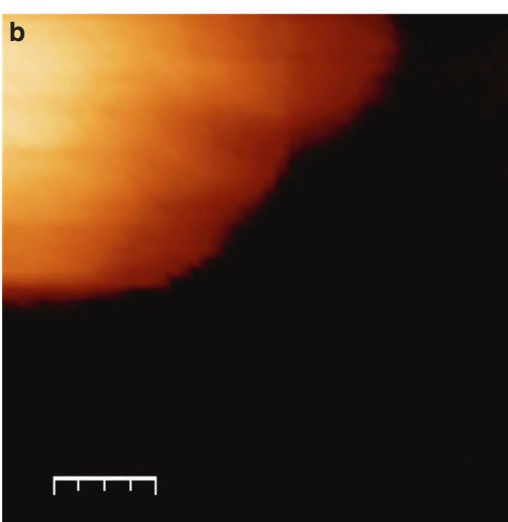

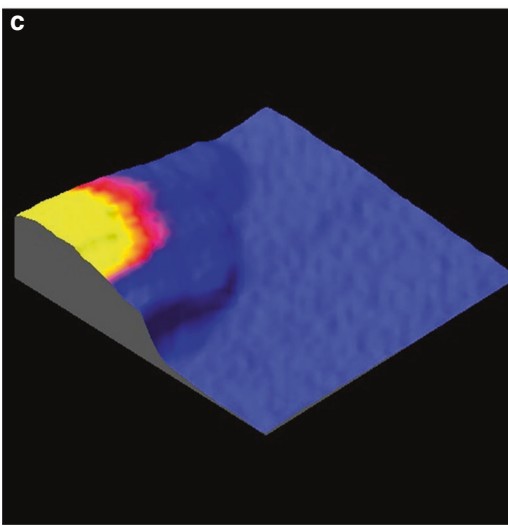

**Fig. 5** Growth of a stable Cu nucleus. Initial (**a**) and final (**b**) images of the growth of a stable Cu nucleus on the ITO substrate over 57 s; the *scale bars* represent 70 nm. Image **c** corresponds to a colour-scale representation of the initial image (*yellow-pink* denotes the dimensions and position of the original nucleus) superimposed on a 3D surface representation of the final image, as seen after 57 s. Supplementary Movie 2 shows the temporal evolution of the particle growth

within the framework of the Scharifker-Hills model[8, 44] are characterized by the so-called instantaneous nucleation model at overpotentials more negative than −0.27 V (Supplementary Fig. 2). The current transients deviate from the "instantaneous" limit towards the "progressive" nucleation at lower overpotentials. Quantitative fitting of the current transients using the Scharifker-Mostany model[45], as exemplified in Supplementary Fig. 3, revealed that the number density of nuclei decreases from approximately $13 \times 10^6$ to $3 \times 10^6$ cm$^{-2}$ upon decreasing the overpotential from −0.29 to −0.24 V. The nucleation rate also shows an increase from 1 to 45 s$^{-1}$ with increasing overpotentials. Similar values were also obtained with the Heerman-Tarallo model[9]. To provide a comparison with other electrode surfaces, similar analysis performed on glassy carbon electrodes show significantly larger currents, with nuclei number densities an order of magnitude higher than on ITO.

Figure 2 illustrates the topography of ITO electrodes after Cu deposition at −0.34 V for 15, 30 and 60 s employing conventional ex-situ AFM. The nucleation was stopped at various times by taking the electrode out of the solution at the applied potential. Over 10 different ITO electrodes were investigated, sampling different areas and different deposition times. From the topographic analysis, nuclei number densities of the order of $(1.5 \pm 0.8) \times 10^8$ cm$^{-2}$ were estimated. We do not observe statistically relevant changes in the nuclei number density with deposition time, which further confirms an "instantaneous" nucleation regime. Analysis of the chronoamperograms employing the Scharifler-Mostany and Heerman-Tarallo models resulted in about one order of magnitude smaller nuclei number density than those observed by AFM. This level of discrepancy has been observed in previous studies[14, 18]. With regards to the nucleation rate, values in the range of 80 s$^{-1}$ were obtained from both models. This nucleation rate is expected to be about one order of magnitude faster than the current spatio-temporal resolution of the HS-LMFM system. In order to decrease the nucleation rate, the concentration of Cu in the electrolytic bath was decreased by an order of magnitude to $1.0 \times 10^{-4}$ mol × dm$^{-3}$. The chronoamperometric transients in Supplementary Fig. 4 show that the nucleation process occurs under a different kinetic regime at this low concentration. The absence of clearly defined maxima and diffusion limiting currents strongly suggest a convolution of the dynamics of electron transfer and nucleation. The physics associated with nucleation mechanisms in this regime are significantly less understood given the weak dependence of the current with time.

**Visualization of Cu nucleation processes via HS-LMFM**. Figure 3 shows a current transient recorded at −0.27 V in $1.0 \times 10^{-4}$ mol × dm$^{-3}$ CuSO$_4$ in the LMFM electrochemical cell, along with snapshot images recorded by in-situ HS-LMFM at two frames per second, revealing the birth of a single stable Cu nucleus. The first frame shows the topographic features associated with the substrate hydration layer at 14 s after the deposition potential is applied. The subsequent frames show the appearance of a stable nucleus over a period of 8 s. Induction times in metal nucleation are characteristic of stochastic process associated with low nucleation rates[31]. This behaviour is more consistent with a 'progressive' nucleation mechanism. Topographic features of the electrode after deposition under identical conditions recorded by ex-situ AFM are displayed in Fig. 3c. The most prominent features are associated with Cu nuclei (as identified by electrostatic force microscopy) with sizes in the range of 6 to 7 nm. The nuclei number density and spatial distribution are compatible with the events recorded by HS-LMFM, yielding a 55% probability of finding a nucleus in 85 × 85 nm$^2$ scan areas. These results can be taken as evidence that the interaction of the VOP with the

hydration layer in the range of 7 nm does not affect the probability of nucleation under these conditions. Control experiments also confirmed that nanometric metallic nuclei do not affect the evanescent field in the optical feedback.

The highly dynamic nature of the process associated with the stochastic birth of a stable nucleus can be seen in the Supplementary Movie 1. A Cu nucleus is formed after approximately 10 s, undergoing partial dissolution before growing into a more stable particle a few nanometres away from the initial nucleation site. After a further 10 to 15 s, the images show the growth of other nuclei in the vicinity of the first nucleus, leading to the formation of a deposit of more than 50 nm in size after a complex sequence of dissolutions/aggregation steps. To the best of our knowledge, this is the first in-situ observation of a single stochastic 3D electrochemical nucleation event in real-time.

The evolution of a copper deposit on a $58 \times 58$ nm$^2$ scan area over a period of 49 s is displayed in Fig. 4. The initial frame in Fig. 4a is taken at 14 s, when a Cu nucleus was already formed at the surface (blue contour). Figure 4b shows the nuclei contour at 14, 22, 32 and 49 s, showing a lateral displacement upon growth. Figure 4c illustrates the evolution of the shear-force, calculated as an additional spring upon the cantilever, across the dotted line drawn in Fig. 4b. It is important to emphasize that the z-scale in these images corresponds to the shear-force of the evolving hydration layers, rather than the height of the nuclei. Analysis of the dynamic HS-LMFM responses suggests that stable nuclei are formed after reaching cross sections of 6 to 7 nm. Smaller clusters tend to either re-dissolve or to 'displace laterally' to aggregate with larger nuclei. Considering the substrate roughness and the separation from it maintained by the optical set point of the VOP, transient features of less than 1 nm may not always be detected. However, stable particles in that size range will affect the hydration layers and therefore be detectable. Consequently, the estimation of stable cluster sizes is well within the topographic resolution of the system. Based on thermodynamic arguments (see Supplementary Fig. 5), the critical nucleus diameter for a hemispherical Cu deposit is approximately 1.5 nm at $\eta = -0.27$ V. Considering that form factors associated with the VOP are not considered in the image analysis, the observed dimensions of stable nuclei bode well with the thermodynamic limit.

The oscillatory nature of electrochemical nucleation has been discussed in the literature and has been linked to branching processes and fractal growth[46]. Previous studies have postulated this type of phenomenon based on ex-situ observations of the shape, size and distribution of metal deposits[47, 48]. Recent studies have concluded that the early stages of nuclei growth involve self-limited growth followed by electrochemically induced surface diffusion, leading to aggregation and coalescence[15–17, 49]. The dynamic features revealed by the HS-LMFM open a window to visualizing these complex local stochastic processes which cannot be extracted from current transients.

Finally, the growth of a stable nucleus is shown in real time in the Supplementary Movie 2. Two snapshots with a 57 s delay are shown in Figs. 5a, b, while Fig. 5c shows a 3D superposition of the nucleus cross-section during the growth transient. The data show a linear increase in the particle cross section with time, at a rate of 323 nm$^2$ s$^{-1}$. Interestingly, extrapolating this linear relationship provides an intercept of approximately 50 s, which coincides with the pre-deposition time used for generating the stable Cu nucleus. Assuming a fully diffusional hemispherical growth, the cross-sectional growth rate would be expected to be approximately $3 \times 10^3$ nm$^2$ s$^{-1}$ [18]. The slower growth rate measured experimentally is a manifestation of the mixed electron transfer/diffusion control kinetics, although contributions from shielding effects cannot be entirely excluded in this length scale.

## Discussion

Real-time monitoring of stochastic electrochemical nucleation events has been achieved with nanometre resolution employing high-speed lateral molecular force microscopy featuring vertically-oriented probes. Adjusting the Cu$^{2+}$ concentration and deposition potential allowed us to establish a regime characterized by stochastic nucleation at ITO electrodes with no more than one nucleus in 100 nm$^2$ on average. Under such a slow deposition rate, the intensity of the evanescent wave, which controls the set-point, is not dampened throughout the deposition transient. The unique aspect of this approach is the ability to measure changes in topography by monitoring fluctuations in the visco-elastic properties of hydration layers. This allows the detection of single nucleation events in real time by scanning the probe at a rate of two frames per second. Sub-second image acquisition unveiled a highly dynamic environment prior to the formation of stable nuclei, featuring nucleation/dissolution events as well as growth via two-dimensional aggregation process. Quantitative measurements of growth rates could be performed on individual nuclei, in one of the most challenging environments to standard probe microscopy, illustrating the potential of this technique to study a wide range of electrochemical processes with high spatiotemporal resolution. In this respect, optically transparent electrodes with lower surface roughness, e.g. graphene layers, would provide the ultimate substrate for this approach.

## Methods

**Lateral molecular force microscopy**. Technical description of the HS-LMFM has been reported elsewhere[38, 40, 41, 50]. Silicon nitride VOPs with resonance frequencies of 380 and 69 kHz, in air and liquid, respectively, and a spring constant of 0.047 N/m were specifically designed for the HS-LMFM (NuNano Bristol). The optical feedback system does not adjust the tip to surface distance to account for sub-wavelength samples, but rather maintains the tip at a constant separation from the substrate from which the evanescent field is generated. The measured images therefore consist of shear-force mapping of the surface at constant height. The interaction range of the VOP has been calibrated to be a 2 nm short range shear-force and 7 nm long range hydrodynamic interaction[50, 51]. Using the intensity of the scattered evanescent field from the VOP tip as a set-point allows the height of the "scanning plane" from the substrate to be raised and lowered, much like the focal plane of an optical microscope. In this way, there is no physical contact between the tip and the substrate. Image processing was performed using a combination of LabView script and WSXM.

**Electrochemical measurements**. The electrolytic solution consisted of an aqueous 50 mM Na$_2$SO$_4$ solution at pH 3, with CuSO$_4$ as the source of copper ions. This electrolytic system was specifically chosen to avoid Cu$^+$ formation, simplifying the analysis of the electrochemical responses[52]. High purity chemicals and ultrapure water (Milli Q system) were employed in all experiments. Electrochemical measurements were performed with an Ivium CompactStat using IviumSoft software. The ITO coated glass used as optically transparent electrode was purchased from SPI Supplies.

**Data availability**. The authors declare that all data supporting the findings of this study are available within the article, the Supplementary Information files or in the University of Bristol Data Repository (data.bris.ac.uk/data) with the identifier DOI:10.5523/bris.18pzsbftq55ky2d2u8vz7syiue.

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

## Acknowledgements

D.P. and D.J.F. are indebted to the EPSRC for financial support (EP/K007025/1 and EP/H046305). K.A.B. acknowledged the support from the EPSRC Centre for Doctoral Training in Functional Materials: The BCFN (EP/L016648/1). The authors gratefully acknowledge Dr. S. Carreira (University of Bristol) and Dr. J.J.L. Humphrey for their support in the early stages of the project, and also thank Dr. A. Antognozzi for constructive discussions. M.J.M. acknowledges a Royal Society Wolfson Merit Award. D.J.F. is also grateful to the Research Fellowship from the Institute of Advanced Studies of the University of Bristol. Support from the University of Bristol Centre for Nanoscience and Quantum Information, Chemistry Imaging Facility and the School of Physics are also acknowledged.

## Author contributions

In-situ microscopy electrodeposition studies were performed by K.A.B., G.H.C., R.L.H. and D.P. K.A.B. designed the in-situ electrochemical cell and performed preliminary testing of the system with R.L.H., who designed and built the HS-LMFM. R.L.H. processed HS-LMFM data to produce movies and diagrams in close consultation with D.P. Ex-situ electrochemical experiments and analysis were carried out by G.H.C. and D.P., while K.A.B., G.H.C. and R.L.H. performed ex-situ microscopy studies. M.J.M. and D.J.F. were responsible for project planning and supervision. All authors discussed the results and analysis. D.P., R.L.H. and D.J.F. co-wrote the manuscript, with contributions from all authors.

## Additional information

**Competing interests:** The authors declare no competing financial interests.

