## [Peer Review File · Nature Communications]

Reviewers' comments:

Reviewer #1 (Remarks to the Author):

The nucleation and growth of metals on electrode surfaces has been researched for decades, but recent studies have introduced major new concepts and perspectives concerning non-traditional nucleation and growth pathways, particularly involving the nucleation and aggregation of clusters to form larger structures, and evidence for the loss (detachment, dissolution) of clusters. Although liquid-phase TEM is advancing apace, the most compelling studies have tended to involve electrochemical measurements coupled with high resolution ex-situ TEM, which does not provide direct information. The present study is highly innovative because it involves the application of in-situ high speed force microscopy of an electrode, under electrochemical control, and reveals important aspects to nucleation and growth, reinforcing and advancing these emerging concepts. As described clearly in the abstract, the results show: "a highly dynamic topographic environment prior to the formation of critical nuclei...[with] formation/re-dissolution of nuclei, 2D aggregation and nuclei growth." This is very important work of high fundamental and applied value.

The paper is very well-written and will be of great interest not just to electrochemists, but to scientists studying interfacial processes generally. It will have significant impact in electrochemistry and encourage the wider use of high speed force microscopy. I just have a few technical questions that could usefully be addressed.

- The method requires an optically transparent substrate and in this case ITO was used.
 - (a) Perhaps add a statement about other substrates that could be used, e.g. graphene on glass among others, to highlight wider applicability of the method?
 - (b) What is the roughness of the ITO and does this effect the minimum cluster size that can be detected? It would be particularly useful to know whether the stable cluster size detected of 6-7 nm, rather than the 1.5 nm calculated for a stable cluster, is impacted in any way by resolution or substrate roughness issues.
- What precautions are taken to avoid thermal drift of the piezos? Is this is an issue for tracking cluster/particle mobility? I appreciate that thermal drift will be less of an issue in high speed measurements, but some information would be useful.
- Likewise, although the probe-surface interaction forces are low, metal NPs can be notoriously mobile and easily detach from surfaces and it would be good to have some more information on the extent to which the tip influences, or not, the reaction environment.
- In terms of purely electrochemical measurements, by working with small scale systems, one can obtain a lot of information on nucleation and growth of individual particles, for example in the classical work of Peter, Fleischmann and others using microelectrode substrates, Mirkin's recent work on electrodeposition at nanoelectrodes coupled with in-situ AFM, recent SECCM studies of Ag and Pd electrodeposition. I am not asking the authors to add references here, but it may be useful to explain that complexity often arises when one has macroscopic systems, although some of these single entity measurements also clearly highlight complexity, such as the importance of cluster aggregation, even in the formation of single nanoparticles, and particle detachment.

Pat Unwin
University of Warwick

Reviewer #2 (Remarks to the Author):

The manuscript describes the visualization of the early stages of Cu electrodeposition on a planar ITO working electrode. This is done using High Speed Lateral Molecular Force Microscopy, which

has the benefit of not disturbing the structure of metal nuclei. What is recorded is a probe frequency shift, which can be converted to shear force.

The approach is novel in visualizing a notoriously difficult to probe phenomenon. The supplemental videos compellingly demonstrate the evolution of shear stress around growing nuclei. The observation of nuclei dissolution and lateral movement is significant.

However, I believe the results should be made more mathematically concrete, which will ultimately result in an improved manuscript. Detailed comments below:

1. Since what is recorded is not the exact metal phase boundary but the shear stress of the dynamic hydration layer, can more detail be provided on how nuclei size can be calculated? Lines 198-200 state an analysis shows the nuclei are stable after achieving 6-7 nm. Can the authors comment on how is this done? Since the metal phase must displace electrolyte, can shear be correlated to volume of metal directly?
- 2, The hydration layers drawn in Fig 1a and 1c show several layers. In data such as Fig 3a, how many of these layers are included in the surface shown?
3. As the technique cannot be performed at $1.0 \times 10^{-3} \text{ mol} \times \text{dm}^{-3}$ concentration, would it not be more appropriate to show the AFM analysis at $1.0 \times 10^{-4} \text{ mol} \times \text{dm}^{-3}$? This would connect more directly to the HS-LMFM results.
4. The equation under Fig S5 has some characters missing that should be fixed. Is the discrepancy between 1.5 nm and 6-7 nm nuclei possibly because portions of the hydration layer are included in that size?

Reference: **NCOMMS-17-04330**

Title: *Real-Time Visualization and Tracking of the Hydrodynamic Signature of Metal Nucleation via Lateral Molecular Force Microscopy*

Ms. Authors: *R.L. Harniman, D. Plana, G.H. Carter, K.A. Bradley, M.J. Miles and D.J. Fermín*

Reply to reviewers

Reviewer #1:

The nucleation and growth of metals on electrode surfaces has been researched for decades, but recent studies have introduced major new concepts and perspectives concerning non-traditional nucleation and growth pathways, particularly involving the nucleation and aggregation of clusters to form larger structures, and evidence for the loss (detachment, dissolution) of clusters. Although liquid-phase TEM is advancing apace, the most compelling studies have tended to involve electrochemical measurements coupled with high resolution ex-situ TEM, which does not provide direct information. The present study is highly innovative because it involves the application of in-situ high speed force microscopy of an electrode, under electrochemical control, and reveals important aspects to nucleation and growth, reinforcing and advancing these emerging concepts. As described clearly in the abstract, the results show: "a highly dynamic topographic environment prior to the formation of critical nuclei...[with] formation/re-dissolution of nuclei, 2D aggregation and nuclei growth." This is very important work of high fundamental and applied value.

The paper is very well-written and will be of great interest not just to electrochemists, but to scientists studying interfacial processes generally. It will have significant impact in electrochemistry and encourage the wider use of high speed force microscopy. I just have a few technical questions that could usefully be addressed.

- The method requires an optically transparent substrate and in this case ITO was used.

(a) Perhaps add a statement about other substrates that could be used, e.g. graphene on glass among others, to highlight wider applicability of the method?

Reply – Indeed, this technique relies on having an electrode which has a low absorption at the wavelength of the TIR laser. The decay length of the evanescent field is critical to the feedback control of the probe distance from the substrate. Closely related to this point, we would also anticipate that the spatial resolution could be compromised by topographic features exceeding the wavelength used to generate the non-propagating evanescent field. ITO has a low surface roughness and is suitably transparent in the necessary wavelength. However, graphene could prove an ideal

substrate for this approach, not only due to the small surface roughness but also since the propagation of the evanescent field is independent of the applied potential. We have included this point on pages 5, 6 and 15 of the revised version.

(b) What is the roughness of the ITO and does this effect the minimum cluster size that can be detected? It would be particularly useful to know whether the stable cluster size detected of 6-7 nm, rather than the 1.5 nm calculated for a stable cluster, is impacted in any way by resolution or substrate roughness issues.

Reply – The average rms roughness of the ITO as measured by TM-AFM on bare ITO coated glass was 1 nm, with an average height of 2.9 nm. The scan height was set to just above the roughness level. Therefore, height fluctuations above 1.5 nm can be detected under these conditions. We cannot exclude the possibility that transient features in the range of 1 nm may not be detected due to the inherent roughness of the substrate. However, if the particle's dimension is stable in time, the system is capable of detecting this feature long before it reaches the 6 – 7 nm stable structures that we observed. Consequently, the 6 to 7 nm stability limit is not dictated by instrument sensitivity or the substrate roughness. We have included these important points on pages 6 and 12 of the revised version.

- What precautions are taken to avoid thermal drift of the piezos? Is this is an issue for tracking cluster/particle mobility? I appreciate that thermal drift will be less of an issue in high speed measurements, but some information would be useful.

Reply –The X-Y scan stage is a commercial system with a quoted resolution of 0.2 nm in each axis. Internal capacitance sensors were also used to measure the location of the scan window. Drift in the z-axis can be neglected by virtue of the feedback control being fully optical. Consequently, the tip-surface distance is independent of the piezos. We have incorporated this information on page 6 of the revised version.

- Likewise, although the probe-surface interaction forces are low, metal NPs can be notoriously mobile and easily detach from surfaces and it would be good to have some more information on the extent to which the tip influences, or not, the reaction environment.

Reply – The force applied normal to the substrate is vanishingly small as the cantilever interacts with the hydration layers. This can be illustrated in the revised figure 1, showing a sequence of steps in the oscillation amplitude of the probe with distance. This represents the first three hydration layers. With the small oscillation of the tip used, the maximum force that the tip can produce parallel to the plane of the surface is approximately 20 pN. This point is clarified on page 6 of the revised version.

- In terms of purely electrochemical measurements, by working with small scale systems, one can obtain a lot of information on nucleation and growth of individual particles, for example in the classical work of Peter, Fleischmann and others using microelectrode substrates, Mirkin's recent work on electrodeposition at nanoelectrodes coupled with in-situ AFM, recent SECCM studies of Ag and Pd electrodeposition. I am not asking the authors to add references here, but it may be useful to explain that complexity often arises when one has macroscopic systems, although some of these single entity measurements also clearly highlight complexity, such as the importance of cluster aggregation, even in the formation of single nanoparticles, and particle detachment.

Reply – This is a very important point that we have expanded in the revised version. We fully agree that confining nucleation to very small areas enables the resolution of single events. In the revised version, we have made explicit reference to phenomena such as induction times, often observed at nanoelectrodes (see for instance ref. 20). We have also made reference to the unique capability of SECCM to investigate the nature of nucleation sites as well as complex processes such as particle nucleation and detachment (see ref. 36). We have included these points on pages 3 and 4 of the revised version.

Reviewer #2:

The manuscript describes the visualization of the early stages of Cu electrodeposition on a planar ITO working electrode. This is done using High Speed Lateral Molecular Force Microscopy, which has the benefit of not disturbing the structure of metal nuclei. What is recorded is a probe frequency shift, which can be converted to shear force.

The approach is novel in visualizing a notoriously difficult to probe phenomenon. The supplemental videos compellingly demonstrate the evolution of shear stress around growing nuclei. The observation of nuclei dissolution and lateral movement is significant.

However, I believe the results should be made more mathematically concrete, which will ultimately result in an improved manuscript. Detailed comments below:

1. Since what is recorded is not the exact metal phase boundary but the shear stress of the dynamic hydration layer, can more detail be provided on how nuclei size can be calculated? Lines 198-200 state an analysis shows the nuclei are stable after achieving 6-7 nm. Can the authors comment on how is this done? Since the metal phase must displace electrolyte, can shear be correlated to volume of metal directly?

Reply – We have modified figure 1 to address this very important point. Figure 1d shows the amplitude of probe modulation as a function of distance from a mica substrate in ultra-pure water. The steps observed in the curves correspond to the various hydration layers, of which there are typically 3 to 6. Each hydration layer is in the region of 2.5 angstroms thick. Under the conditions of the nucleation experiments, we estimate that the probe is located approximately in the second or third hydration layer, which results in an uncertainty of less than 1 nm in the detection. Consequently, the value of 6 to 7 nm quoted for stable nuclei is well within the detection range of the instrument. This point is discussed on pages 5, 6 and 12 of the revised version.

The scanning plane is raised to the level that the basic roughness of the ITO is just at the edge of the observable range. Therefore, the hydrodynamic signature and pattern of growth should be observable until the height of the nano-particle is almost in physical contact with the tip. This should mean that no growth is missed as even sub-nanometre features should produce a hydrodynamic signature in the observable height range. These comments have been included on page 6 and 12 of the revised version.

The suggestion of correlating shear to volume is interesting and there is currently an ongoing project to use sliding-mode observer algorithms to achieve something along these lines. However, in the current study the level of the shear measured is only qualitatively linked to the height of the observed object. The “scanning plane” is held at a set distance above the ITO and the shear force experienced by the tip increases as the top of a growing object beneath gets closer to the tip. However, the direct correlation between shear measured and height of the object requires further investigation. For instance, we presently intend to investigate the specific influence of applied potentials on the structure of individual hydration layers.

2, The hydration layers drawn in Fig 1a and 1c show several layers. In data such as Fig 3a, how many of these layers are included in the surface shown?

Reply – As mentioned above, figure 1d has been included to illustrate the number of hydration layers typically recorded at an oxide surface (mica). Under the electrochemical environment, we estimate that the probe interacts with the second or third hydration layer (page 6 of the revised version)

3. As the technique cannot be performed at 1.0×10^{-3} mol \times dm $^{-3}$ concentration, would it not be more appropriate to show the AFM analysis at 1.0×10^{-4} mol \times dm $^{-3}$? This would connect more directly to the HS-LMFM results.

Reply – The conventional AFM analysis provided a clear trend in the fast nucleation regime (CuSO_4 $1.0 \times 10^{-3} \text{ mol} \times \text{dm}^{-3}$), as the size of the nuclei enabled a clear topographic contrast with the substrate (see figure 2). On the contrary, ex-situ nuclei detection below 60 s in the case of $1.0 \times 10^{-4} \text{ mol} \times \text{dm}^{-3}$ was somewhat compromised by the contrast with the substrate topography. For this reason, we decided to show a representative ex-situ image recorded at 60 s. In order to close the gap between the two concentration regimes, we analysed chronoamperometric steps with a concentration of $5.0 \times 10^{-4} \text{ mol} \times \text{dm}^{-3}$, employing the Scharifker-Mostany model. The analysis shows an overall decrease in nuclei density and nucleation rates with respect to the $1.0 \times 10^{-3} \text{ mol} \times \text{dm}^{-3}$ electrolyte case. Although a semi-quantitative trend can be observed in terms of nucleation rates, more experiments are required. In particular, decreasing even further the substrate roughness would substantially enhance detection limits in the HS-LMFM and conventional AFM methods.

4. The equation under Fig S5 has some characters missing that should be fixed. Is the discrepancy between 1.5 nm and 6-7 nm nuclei possibly because portions of the hydration layer are included in that size?

Reply – We apologise for the missing characters. Based on the example included in the revised version of figure 1, the probe is estimated to be interacting with the second or third hydration layer, which corresponds to distance of approximately 1 nm. However, the origin of the discrepancy lies on the overlap between the VOP shape and the hydration layer. This is the point we raised in the original submission (see page 12).

REVIEWERS' COMMENTS:

Reviewer #1 (Remarks to the Author):

What was already an excellent paper has been further improved through this thorough review. This is very important work of high quality and should be published.

Reviewer #2 (Remarks to the Author):

Thank you for the replies. I believe the expanded explanation of the technique contributes greatly to clarity, and makes the work more accessible to a broad audience. Figure 1 (d) makes the connection quite clear.

I recommend labelling the blue gradient in Fig 1(a) as "3 hydration layers" to avoid any misinterpretation by the reader.

Reference: NCOMMS-17-04330

Title: **Real-Time Tracking of Metal Nucleation via Local Perturbation of Hydration Layers (Revised)**

Authors: *R.L. Harniman, D. Plana, G.H. Carter, K.A. Bradley, M.J. Miles & D.J. Fermín*

REVIEWERS' COMMENTS

Reviewer #1 (Remarks to the Author):

What was already an excellent paper has been further improved through this thorough review. This is very important work of high quality and should be published.

Reviewer #2 (Remarks to the Author):

Thank you for the replies. I believe the expanded explanation of the technique contributes greatly to clarity, and makes the work more accessible to a broad audience. Figure 1 (d) makes the connection quite clear.

I recommend labeling the blue gradient in Fig 1(a) as "3 hydration layers" to avoid any mis-interpretation by the reader

We are delighted by the positive assessments by both reviewers. We have implemented the modification in the label of Figure 1a as suggested by Reviewer 2. We very much appreciate the comments by both reviewers, which have improved the clarity of the paper.